# Molecular determinants of permeation in a fluoride-specific ion channel

Nicholas B Last, Senmiao Sun, Minh C Pham, Christopher Miller*

Department of Biochemistry, Howard Hughes Medical Institute, Brandeis University, Waltham, United States

**Abstract** Fluoride ion channels of the Fluc family combat toxicity arising from accumulation of environmental $F^-$. Although crystal structures are known, the densely packed pore region has precluded delineation of the ion pathway. Here we chart out the Fluc pore and characterize its chemical requirements for transport. A ladder of H-bond donating residues creates a 'polar track' demarking the ion-conduction pathway. Surprisingly, while track polarity is well conserved, polarity is nonetheless functionally dispensable at several positions. A threonine at one end of the pore engages in vital interactions through its β-branched methyl group. Two critical central phenylalanines that directly coordinate $F^-$ through a quadrupolar-ion interaction cannot be functionally substituted by aromatic, non-polar, or polar sidechains. The only functional replacement is methionine, which coordinates $F^-$ through its partially positive γ-methylene in mimicry of phenylalanine's quadrupolar interaction. These results demonstrate the unusual chemical requirements for selectively transporting the strongly H-bonding $F^-$ anion.

DOI: https://doi.org/10.7554/eLife.31259.001

## Introduction

Aqueous fluoride ion is an omnipresent environmental xenobiotic that inhibits certain phosphoryl-transfer enzymes critical for energy production and nucleic acid synthesis (*Marquis et al., 2003*; *Samygina et al., 2007*). Two phylogenetically unrelated $F^-$ efflux systems are now known that resist the toxicity of cellular $F^-$: the $CLC^F$ family of $F^-/H^+$ antiporters, and the Fluc family of $F^-$ ion channels (*Baker et al., 2012*; *Stockbridge et al., 2012*). Fluc channels, present in all classes of life except higher animals, allow passive transit of $F^-$ ions across the membrane down their electrochemical gradient. Bacterial Fluc channels are constructed as small, dual-topology dimers; the two subunits, each of ~120 residues, span the membrane in opposite orientations to form the functional complex (*Stockbridge et al., 2013*). These channels are notable for this unusual architecture as well as for exceptionally high $F^-/Cl^-$ selectivity. Recent crystal structures of two Fluc channels (*Stockbridge et al., 2015*; *Last et al., 2016*) reveal four bound $F^-$ ions whose transmembrane disposition imply two antiparallel ion-permeation pathways, each occupied simultaneously by two $F^-$ ions (*Figure 1a*, *Figure 1—figure supplement 1*). Each pathway was postulated to be extremely narrow, largely anhydrous, and filled up by side chains that hand off $F^-$ ions passing along it (*Figure 1b*). This unusual structural inference was confirmed functionally by substitutions of three conserved, ion-coordinating residues in this region – two phenylalanines and an asparagine – that fully inhibit $F^-$ transport.

Zooming in on a single pore of the Fluc homologue under study here (*Figure 1b*), we see two bound $F^-$ ions separated by 12 Å and surrounded mainly by protein side chains. The lower ion approaches one of the wide aqueous vestibules on the two ends of the channel, while the other is deeply buried within the protein. The proposed critical pore-associated moieties are of two chemical types: aromatic rings of two conserved Phe residues contributed from different subunits, each of which coordinates a $F^-$ ion in an electropositive edge-on orientation, and dipolar H-bond donors

*For correspondence:
cmiller@brandeis.edu

Competing interests: The authors declare that no competing interests exist.

arranged to satisfy the F⁻ ion's H-bond accepting preference. These H-bonding residues were proposed (*Stockbridge et al., 2015*) to form a 'polar track' extending in both directions beyond the centrally located Phe rings, as indicated in the sequence alignment and structure of *Figure 1*. Most polar track residues emanate from TM4, where they appear at every 4th position, and although not all are individually conserved, H-bond donors frequently appear here. Several of these coordinate F⁻ ions in the crystal structures, while others lie close to the aqueous vestibules where crystallographically ordered waters are also observed. We emphasize that the structures do not show water-filled aqueous tunnels lined by polar groups, as in many other ion channels; instead, the pores are filled with a web of side chains (*Figure 1c*) that must be negotiated by ions translocating at rates on the order of $10^6$–$10^7$ ions/sec.

Our aim here is to further test this picture by mutating each of the residues thought to be involved in F⁻ transport and to determine their chemical requirements for channel function. Results

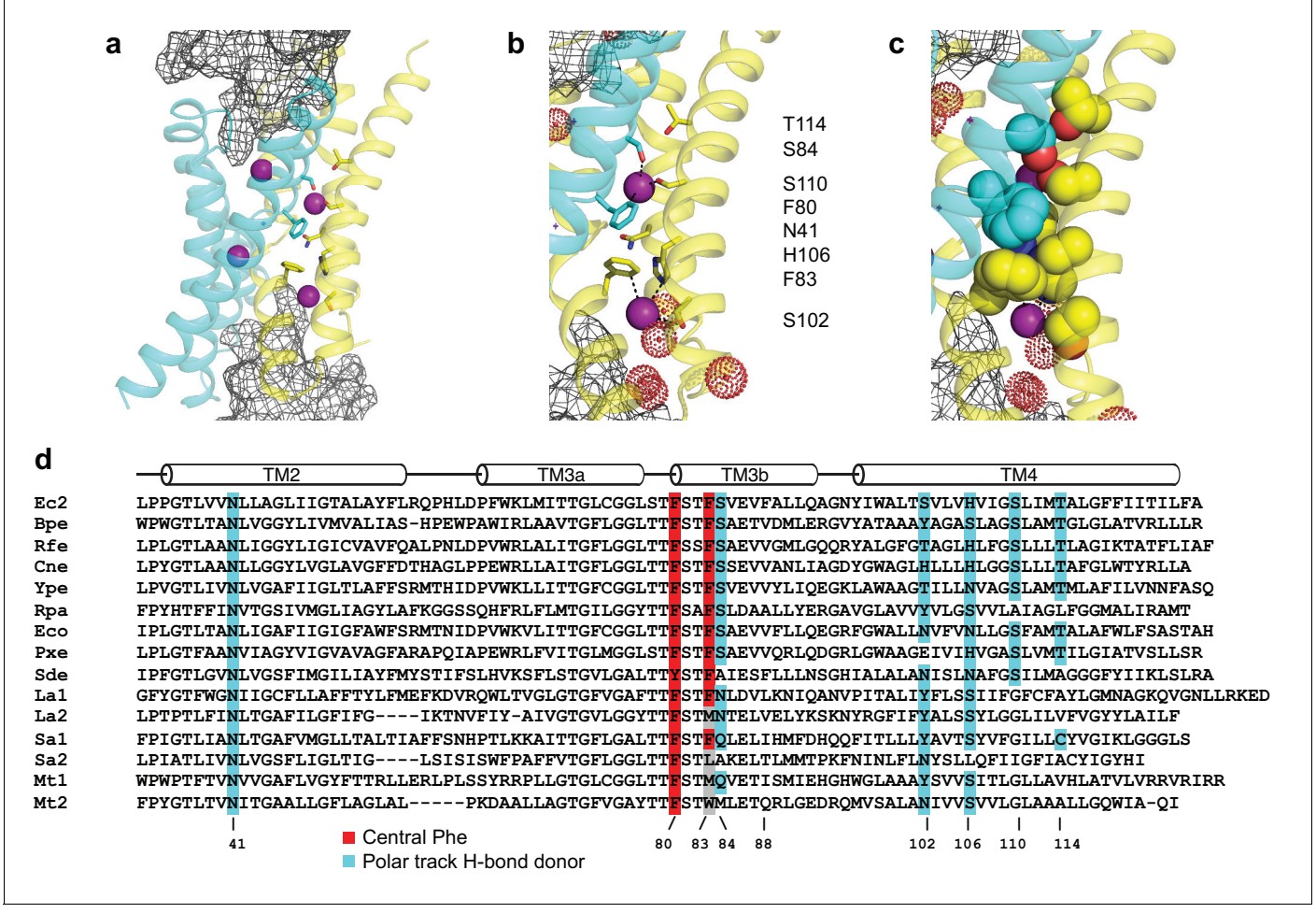

**Figure 1.** Fluc-Ec2 channel structure. (a) Crystal structure of Ec2-S9 complex (PDB #5A43, re-refined) showing double-pore assembly, with F⁻ ions in purple and subunits in cyan and yellow. Grey mesh marks surfaces of the bound S9 monobodies delineating the channel's aqueous vestibules. (b) Blow-up of an individual pore region, with coordinating Phe and polar track side chains indicated and colored according to the subunit from which they project. Crystallographic waters are shown as dotted spheres. (c) Space-fill representation of same region. (d) Bacterial Fluc sequence alignment spanning polar track residues (TM2-TM4), with color code indicated. Top two sequences (Ec2, Bpe) refer to structurally known homodimeric homologues. Known or surmised heterodimers are shown in six lower sequences (La1/2, Sa1/2, Mt1/2). Other sequences are taken arbitrarily from Swissprot for illustration.

DOI: https://doi.org/10.7554/eLife.31259.002

The following figure supplement is available for figure 1:

**Figure supplement 1.** Stereo (wall-eye) view of *Figure 1a*.
DOI: https://doi.org/10.7554/eLife.31259.003

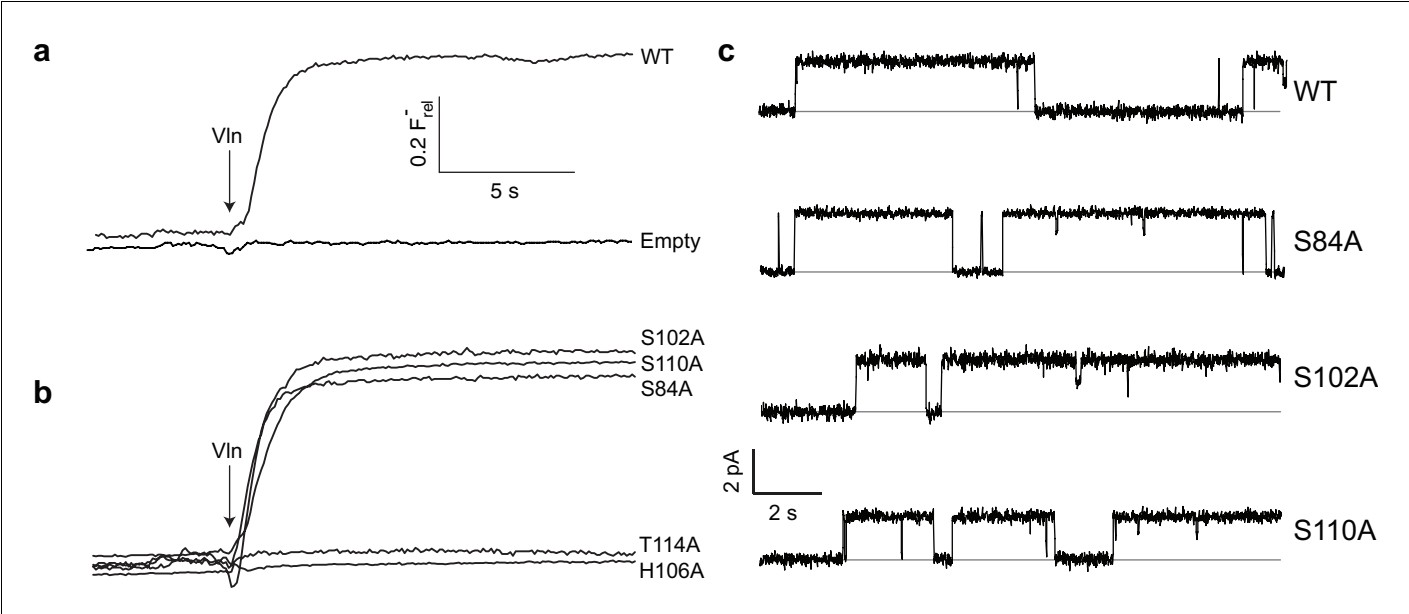

**Figure 2.** Effect of polar track mutation on Fluc function. (**a**) WT fluoride flux, limited by the response time of the measurement. (**b**) Polar track Ala mutations. (**c**) Single channel records of active polar track mutants in the presence of reversible blockers (monobodies) that completely inhibit flux and allow measurement of the zero-current level (gray line).

DOI: https://doi.org/10.7554/eLife.31259.004

The following figure supplements are available for figure 2:

**Figure supplement 1.** F⁻ efflux recordings of N41 and H106 mutants.

DOI: https://doi.org/10.7554/eLife.31259.005

**Figure supplement 2.** Mutants retain impermeability to Cl⁻.

DOI: https://doi.org/10.7554/eLife.31259.006

confirm that the mutation-sensitive positions do indeed trace along the trajectory expected, but we were surprised to find that the polar character of several of these residues is not required for channel activity. Positions that are functionally sensitive to mutagenesis require a specific, finely tuned side-chain, and a conserved Thr residue unexpectedly uses its β-branched methyl group rather than its H-bonding hydroxyl. An additional surprise is that no other aromatic or canonical polar group substituted at the conserved Phe residues supports channel activity, but Met at one of these positions leads to a fully active channel; this seemingly strange result is rationalized by an electrostatic argument that the Met side chain mimics the Phe ring, a prediction confirmed by a crystal structure of the mutant.

## Results

A Fluc homologue from an *E. coli* virulence plasmid, nicknamed 'Ec2' (*Stockbridge et al., 2013*), was used for all experiments. Three strongly conserved, deeply buried residues spanning ~7 Å along the pore were previously found to be required for transport (*Stockbridge et al., 2015*; *Last et al., 2016*) in this and a different homologue: two Phe (F80, F83) and an Asn positioned between them (N41). To initially gauge the functional importance of the entire polar track, we mutated each of the four track residues on TM4 (S102, H106, S110, T114) to Ala as well as an additional residue, S84, which in the structure projects into the pore-region from TM3 of the partner subunit. Channel activity was assessed with an anion-efflux assay, wherein F⁻ or Cl⁻ transport by liposome-reconstituted Fluc was followed continuously with ion-specific electrodes (*Stockbridge et al., 2013*). Because F⁻ flux through WT channels is far faster than the electrode's time-response, this is effectively a binary assay of whether a construct is broadly functional or severely impaired (*Figure 2a*); even a channel with a 95% inhibited transport rate will display a WT phenotype, completely emptying its liposomes of F⁻ within the electrode's response time. For this reason, all mutants appearing active in the flux

**Table 1.** F⁻ transport activity of Ec2 channel mutants.

Activity (single-channel conductance), relative to WT, was calculated for mutants showing efflux behavior similar to WT by recording single-channel currents at 200 mV and symmetrical F⁻, and normalizing to WT current under identical conditions. Mutants scored 'X' gave no discernable flux, equivalent to a relative turnover $<10^{-4}$ of the WT rate. F80I and F83I results are from *Last et al. (2016)*.

| Construct | Activity | Construct | Activity |
|---|---|---|---|
| WT | 1 | N41S | X |
| | | N41Q | X |
| F80A | X | S84A | 1.0 ± 0.1 |
| F80L | X | | |
| F80I | X | F88A | 0.63 ± 0.06 |
| F80Y | X | | |
| F80W | X | S102A | 1.0 ± 0.1 |
| F80S | X | | |
| F80T | X | H106A | X |
| F80Q | X | H106S | X |
| F80H | X | H106N | X |
| F80M | 0.78 ± 0.05 | H106Y | X |
| | | H106W | X |
| F83A | X | H106F | X |
| F83L | X | | |
| F83I | X | S110A | 0.80 ± 0.06 |
| F83Y | X | | |
| F83W | X | T114A | X |
| F83S | X | T114S | X |
| F83H | X | T114V | 0.99 ± 0.06 |
| F83M | X | T114I | 1.1 ± 0.1 |

DOI: https://doi.org/10.7554/eLife.31259.007

assay were also electrophysiologically examined in planar phospholipid bilayers for direct determination of single-channel conductance. Surprisingly, we find (*Figure 2b*) that F⁻ robustly permeates three track mutants denuded of their hydroxyl groups - S84A, S102A, and S110A - with single-channel conductances within 25% of WT (*Figure 2c*, *Table 1*). In contrast, H106A and T114A, as well as alternative H-bonding substitutes H106S, H106Y, H106W, H106N, H106F, N41S, N41Q, and N41Y are completely nonfunctional (*Figure 2b*, *Figure 2—figure supplement 1*), representing >10,000 fold rate-inhibition relative to WT. In crystal structures (*Figure 1b*) H106 coordinates a partially hydrated F⁻ ion located close to bulk solvent at one end of the narrow pore, T114 lies beyond the other F⁻ ion at the opposite end, and the other residues tested reside between them. These functionally sensitive positions thus chart out a ~ 20 Å span connecting the deepest parts of the aqueous vestibules, as suggested from the structures alone. We now turn to the chemical requirements for three of the positions vital for transport.

## F⁻-coordinating aromatics

The two F⁻-coordinating Phe residues stand out due to their unusual anion-quadrupolar coordination and their strong conservation. Despite the residue's hydrophobicity, its π-electrons leave a partial positive charge on the ring hydrogens (*Mecozzi et al., 1996*), such that each ring approaches a F⁻ ion in an edge-on orientation. Removal of aromaticity by substitution of either Phe with Ile was previously shown to abolish both transport and F⁻ occupancy at the site near the substitution (*Last et al., 2016*). We examined a variety of mutations to survey the chemical requirements for F⁻ binding and transport at these central Phe positions. Aromatics (Y, W, H) were introduced as conservative

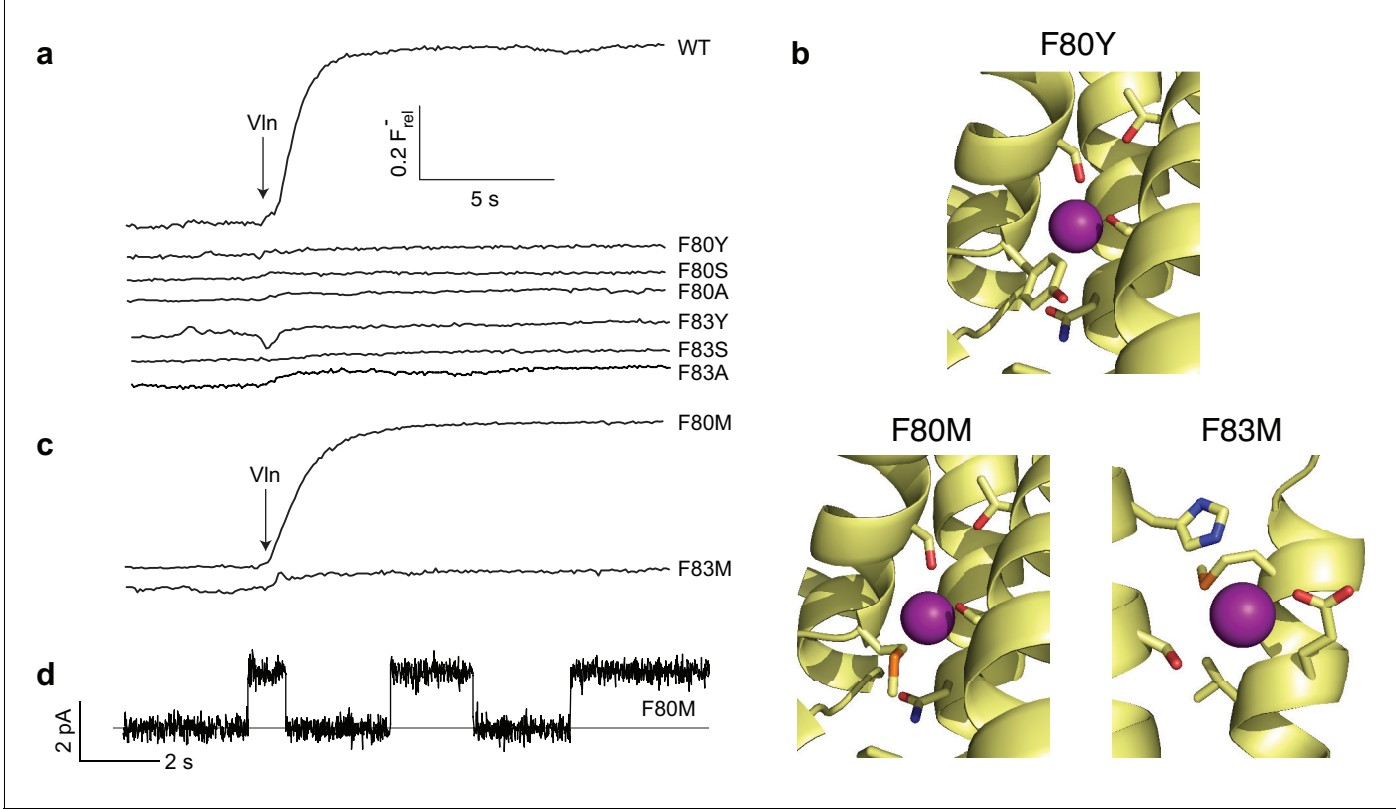

**Figure 3.** Mutagenesis of central Phe residues. (**a**) F⁻ efflux traces of indicated mutants. (**b**) Structures of Phe mutants in F⁻-coordination region. (**c**) F⁻ efflux traces of Met substitutions at central Phe residues. (**d**) Single-channel recording of F80M.

DOI: https://doi.org/10.7554/eLife.31259.008

The following figure supplements are available for figure 3:

**Figure supplement 1.** F- efflux recordings of central Phe mutants.

DOI: https://doi.org/10.7554/eLife.31259.009

**Figure supplement 2.** Stereo views of *Figure 3b*.

DOI: https://doi.org/10.7554/eLife.31259.010

**Figure supplement 3.** Crystal structures of central Phe mutants. F⁻-coordinating region of the indicated mutants (yellow) are shown along with aligned WT showing positions of side chains (cyan) and corresponding F⁻ ion (purple dots).

DOI: https://doi.org/10.7554/eLife.31259.011

**Figure supplement 4.** F⁻ density in central Phe mutants.

DOI: https://doi.org/10.7554/eLife.31259.012

substitutions, aliphatic H-bond donors (S, T, Q) as potential F⁻-coordination partners of alternative chemistry, and hydrophobic aliphatics (L, A) as additional controls. All eight mutations, most of them tested at both Phe residues, completely abolish channel activity (*Figure 3a*, *Figure 3—figure supplement 1*). The most conservative variant, Tyr, is particularly striking since it differs from Phe only by the presence of the ring hydroxyl. A crystal structure of the F80Y mutant, solved to assess F⁻ binding and structure of the pore region, is essentially identical to WT (C$_\alpha$ rmsd ~0.3 Å) with the Tyr ring congruent to that of WT Phe (*Figure 3b*, *Figure 3—figure supplements 2* and *3*). In contrast to previous structures of inactive Ile mutants (*Last et al., 2016*), F⁻ remains bound to the Tyr construct in precisely the same positions as in WT (*Figure 3—figure supplement 4*). Other than the added hydroxyl group, no significant structural changes between WT and the inactive F80Y mutant are apparent, nor does the structure suggest a clear reason why F⁻ fails to permeate the mutant.

While the coordinating Phe residues are strongly conserved among Fluc homologues, they are not perfectly so, as illustrated in the alignment of *Figure 1d*. Bacterial Fluc sequences present a complication that must be appreciated in interpreting alignments. While many Fluc proteins are homodimeric like Ec2, others are known or inferred heterodimers as in *Lactobacillus acidophilus*

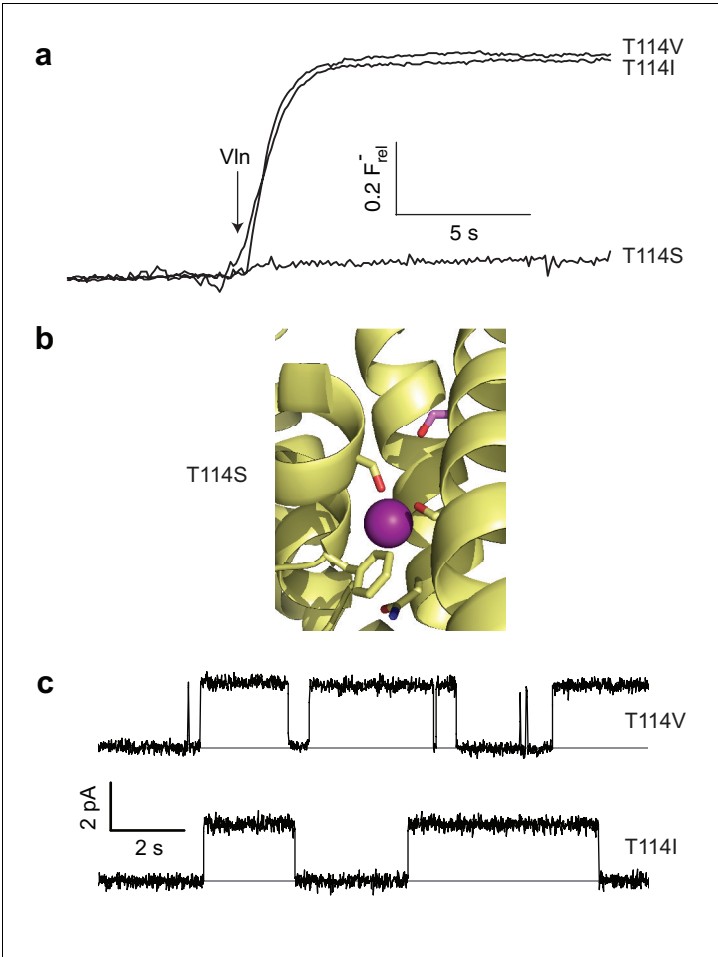

**Figure 4.** Unexpected T114 interaction. (**a**) Active and inactive T114 substitutions in F$^-$ efflux assay. (**b**) F$^-$-coordinating region of T114S crystal structure. Mutated residue is in pink. (**c**) Illustrative single-channel recordings of apolar substitutes scoring active in flux assay.

DOI: https://doi.org/10.7554/eLife.31259.013

The following figure supplements are available for figure 4:

**Figure supplement 1.** Stereo view of *Figure 4b*.
DOI: https://doi.org/10.7554/eLife.31259.014

**Figure supplement 2.** Overlay of WT and T114S structures. Blow-up of F$^-$- coordinating region of T114S crystal structure (yellow) aligned with WT (cyan).
DOI: https://doi.org/10.7554/eLife.31259.015

**Figure supplement 3.** Close contacts with the T114 β-branched methyl.
DOI: https://doi.org/10.7554/eLife.31259.016

**Figure supplement 4.** F$^-$ efflux (**a**) and single-channel recording (**b**) of F88A.
DOI: https://doi.org/10.7554/eLife.31259.017

(La1, La2), *Staphylococcus aureus* (Sa1, Sa2), and *Mycobacterium tuberculosis* (Mt1, Mt2). This circumstance offers the possibility of genetic drift, such that one of the two pores becomes inactivated. As long as the second pore remains active, the inactivated pore can lose residues vital for transport with no physiological effect. In such cases, non-canonical residues appearing in the alignment might either reflect a novel transport chemistry, or simply represent a functionally disrupted pore that no longer has strict residue requirements. Such an arrangement has been suggested from sequence analysis (*Stockbridge et al., 2015*) and indicated experimentally in a yeast Fluc channel (*Smith et al., 2015*; *Berbasova et al., 2017*). Thus, both La and Sa heterodimers retain three of the central phenylalanines, but have either Leu or Met appear at the fourth position. Given that one

pore in these dimers would still possess both phenylalanines, the altered residues may line an inacti-vated pore. The Mt heterodimer surprisingly loses both phenylalanines at the second position, replaced with either methionine or tryptophan. This leads to the unexpected hypothesis that that either Met or Trp are capable of coordinating and transporting fluoride in that construct.

Since alignments show Met occasionally appearing at the central Phe positions, we wondered if this might be functionally meaningful, despite ambiguities about heterodimeric sequences. In this context, Met can be viewed as a uniquely plausible non-aromatic substitute for Phe, since the elec-tron-withdrawing sulfur is expected to produce a partial positive charge on the γ-methylene and ter-minal methyl groups, either of which might act as an electrostatic surrogate to coordinate F⁻. Accordingly, each central Phe was mutated to Met and tested for F⁻ transport. Remarkably, F80M results in an active flux-phenotype (*Figure 3c*) and displays clean single-channel recordings with near-WT conductance (*Figure 3d*, *Table 1*). In contrast, F83M is inactive.

We determined crystal structures of both Met mutants to test the prediction above regarding F⁻ coordination. As with the Tyr substitute, the Met structures are virtually identical to WT (*Figure 3b*, *Figure 3—figure supplement 2*), and all residues surrounding the substitutions are unmoved (C$_\alpha$ rmsd ~0.3 Å from WT, *Figure 3—figure supplement 3*). The Met side chains in both take on a twisted conformation, filling approximately the same space as the Phe ring in WT. As predicted, in both Met mutants F⁻ ions are coordinated by the γ-methylenes located in nearly the same positions as the coordinating Phe ring carbons in WT (*Figure 3—figure supplements 3* and *4*), as though the substituted linear side chains attempt to mimic Phe's aromatic ring in occupied volume and F⁻-bind-ing electrostatics. As with the Tyr mutant, the similar structures offer no explanation why F80M is functionally active and F83M is not.

## A paradoxically polar position

As described above, the moderately conserved Thr114 lies at one end of the polar track, distant from bound F⁻ ions, and its substitution by Ala abolishes channel activity, as if its hydroxyl group is necessary for ushering the anion into the narrow pore from the aqueous vestibule. However, a stark refutation of this idea emerges from the complete inactivity of T114S (*Figure 4a*), despite a T114S crystal structure showing the Ser hydroxyl exactly overlaying its WT counterpart and no significant structural changes elsewhere (C$_\alpha$ rmsd ~0.2 Å, *Figure 4b*, *Figure 4—figure supplements 1* and *2*). This perplexing result motivated an examination of Thr's β-branched methyl group as a functional determinant, and indeed Val and Ile were found to be fully active substitutes here in both liposome F⁻ efflux and single-channel recording experiments (*Figure 4a,c*, *Table 1*).

What could be the function of the β-branched methyl group? In the WT structure, this contacts the Gly49 C$_\alpha$, V85 C$_\alpha$, and F88 side chain, all from the partner subunit (*Figure 4—figure supple-ment 3*). Thus, it is possible that the T114 methyl participates in a nonpolar interaction critical for stabilizing the conducting structure of WT (a structure that nevertheless is still observed in T114S crystals). We attempted mutating Phe88 to Ala (mutation of the backbone C$_\alpha$'s being unfeasible) to weaken the interaction with the opposite subunit. The height of the F88A flux kinetic is clearly smaller than that of WT (*Figure 4—figure supplement 4*), demonstrating that a significant fraction of the channels, although apparently well-folded and dimeric off the gel-filtration column, are inac-tive. However, those channels that remain active have near WT levels of single channel flux (*Fig-ure 4—figure supplement 4*, *Table 1*). These results are difficult to interpret, as the changes in the fraction of active F88A may be independent of any possible interaction with T114. Additionally, given that T114 also makes close contact with the F88 beta carbon (which would remain in the F88A construct, *Figure 4—figure supplement 3*), interactions vital for flux may still remain even after mutation. We therefore conjecture, without compelling evidence beyond the crystal structure itself, that Thr114 contributes to the pore's structural integrity but does not coordinate F⁻ during its transit.

A striking feature of Fluc channels is their extreme (>10,000 fold) selectivity against Cl⁻ ions (*Stockbridge et al., 2013*). This may simply reflect steric hindrance arising from a narrow bore of the conduction pathway, but no evidence currently informs this question pro or con. Whatever the mechanism of Cl⁻ exclusion, though, it is preserved in all functional mutants described here; in no case do we observe any Cl⁻ leakage through these channels (*Figure 2—figure supplement 2*), a result further supporting the structural propriety of the mutant channels independent of their crystal structures.

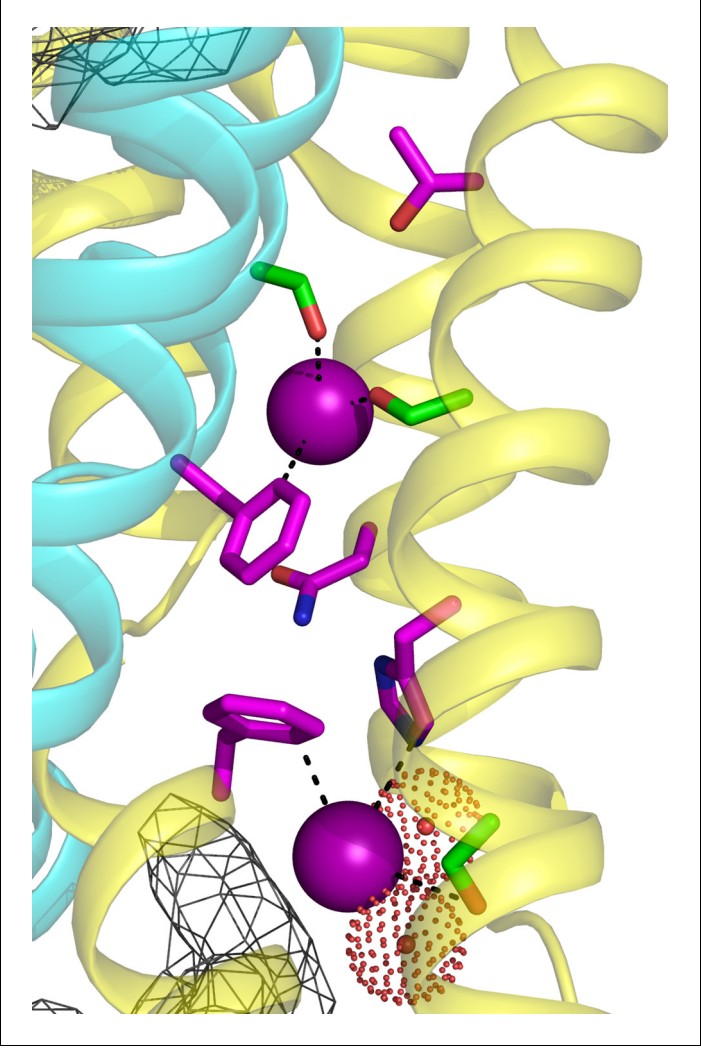

**Figure 5.** Summary of critical pore regions in Fluc Ec2. View of single pore with side chains scored as functionally sensitive to mutagenesis (magenta) or tolerant of mutagenesis (green).

DOI: https://doi.org/10.7554/eLife.31259.018

The following figure supplement is available for figure 5:

**Figure supplement 1.** Stereo view of *Figure 5*.

DOI: https://doi.org/10.7554/eLife.31259.019

## Discussion

Despite functional experiments accompanied by crystal structures at resolution high enough to locate F⁻ ions, revealing two parallel and independent pores, the F⁻ permeation pathway of Fluc channels remains mysterious. The plug of sidechains clogging the pore leaves no clearly visible ion-diffusion pathway, thus provoking our proposal for an unconventional 'channsporter' permeation mechanism (*Stockbridge et al., 2015*) whereby F⁻ passage is accomplished by concomitant movements of side-chain rotamers on a submicrosecond timescale. Experiments here have now delineated the critical residues along the length of the pore (*Figure 5*, *Figure 5—figure supplement 1*); its overall trajectory, from the solvent-accessible binding site at F83/H106 to the terminus near T114, is as suggested from the structures, but several of the new findings are unexpected.

The requirements for the central Phe positions, which stand out prominently in sequence alignments and crystal structures, mirror what is seen in nature: only rare replacements of Phe. The failure of even Tyr to functionally substitute for Phe at either Ec2 position is particularly surprising in light of the F80Y crystal structure. The Tyr and Phe rings and the F⁻ ions adopt identical positions, and the

Tyr hydroxyl points away from the bound F⁻, its presence leaving other parts of the protein undisturbed. Modification of ring electrostatics by the polarizable hydroxyl may lead to a more tightly bound ion, inhibiting flux. If any sidechain movement is required during transport, the additional hydroxyl may introduce either steric clash or competing polar interactions in those alternate conformers. In the case of F80Y, favorable hydroxide-phenyl interactions with the neighboring F83 could also inhibit sidechain movement.

Also surprising is the full activity of the F80M substitution. This may be rationalized by the F⁻-coordinating γ-methylene, whose weakly electropositive character recalls the electropositive, edge-on presentation of the Phe ring to its coordinated F⁻ ion. The Met side-chain configuration in the F80M structure is in striking agreement with this idea, but the functional impairment of F83M, which offers a similar configuration, remains puzzling. F83 coordinates its fluoride in conjunction with H106, which is the only other residue in this construct that will brook no substitution. This raises the possibility that more precise orientation is required here than at F80. While the Ec2 homologue studied here is inflexible at this location, different coordination schemes are clearly possible; *M. tuberculosis* (*Figure 1d*) replaces both the histidine and phenylalanine. It remains to be seen whether these alternate arrangements continue to be as finely tuned and unaccepting of modification.

Other aliphatic mutations at these critical Phe positions are known to eliminate transport (*Figure 3a*, *Figure 3—figure supplement 1*, *Table 1*), and this prohibition now extends to dipolar H-bond donors, with neither Ser, Thr, His, nor Gln supporting channel activity. Previous removal of the central phenylalanines (F80I, F83I) led to a loss of fluoride density in crystal structures (*Last et al., 2016*), suggesting that the F83M and F80Y mutants inhibit transport via a mechanism fundamentally different from the simple loss of a binding site. The results establish that the center of the pore has an exceptionally strict requirement for F⁻ coordination and transport, possibly reflecting a need for a geometrically precise, electrostatically weak binding partner for rapid transit of F⁻ in this anhydrous region.

Along the conserved polar track, only two H-bond donors are required: N41 and H106. The N41 residue, among the most strictly conserved in the Fluc family, is oriented between the two F⁻-coordinating Phe, well positioned to hand off the anion between them. In coordinating a partially hydrated F⁻ at the end of the pore, H106 could catalyze dehydration as the ion leaves bulk water in the vestibule. It is surprising, even alarming, that for the hydroxyl-bearing track residues - S84, S102, S110, and T114 - polarity and H-bonding are not directly involved in anion permeation, despite the conservation of polarity at these positions and their proximity to bound F⁻ ions in the pore. This apparent discrepancy between structural and functional facts presents a mechanistic puzzle calling out for future attack.

## Materials and methods

### Reagents

Chemicals obtained from Sigma-Aldrich (St. Louis, MO) or Fisher Scientific (Waltham, MA) were of highest grade. *E. coli* mixed polar phospholipids (EPL) were from Avanti Polar Lipids (Alabaster, AL), and n-decylmaltoside (DM), n-dodecylmaltoside (DDM), and 3-[(3-Cholamidopropyl)-Dimethylammonio]-1-Propane Sulfonate] (CHAPS) from Anatrace (Maumee, OH). K-isethionate solutions were prepared from isethionic acid (Wako Pure Chemical Industries [Osaka, Japan]) titrated with KOH.

### Protein purification and Liposome reconstitution

The 'wildtype' (WT) Ec2 construct used here carries a C-terminal hexahistidine tag as well as a single mutation (R25K) that enhances protein expression but does not affect function (*Stockbridge et al., 2015*). Mutant Ec2 channels were constructed by standard PCR techniques and were expressed, purified, and reconstituted as previously described (*Last et al., 2016*), as was the S9 monobody. In brief, BL21(DE3) cells were transformed with pET21 vectors containing Fluc constructs. Cells were grown to an OD of 1.5 in terrific broth at 37 C, induced with 2 mM isopropyl β-D-1-thiogalactopyranoside (IPTG) for 1 h at the same temperature, and then harvested by centrifugation. Cell pellets were resuspended in 50 mM Tris, 100 mM NaCl, pH 7.5 with small amounts of DNAse and lysozyme. Cells were broken by sonication, and channels were extracted with 1% DDM for 2 h at RT. Solubilized lysate was clarified by centrifugation, and then loaded on Talon cobalt resin equilibrated with

20 mM Tris, 100 mM NaCl, 5 mM DM, pH 7.5. Cobalt columns were washed and then eluted with 40 mM and 400 mM imidazole, respectively, in equilibration buffer. Cobalt elution was concentrated, and run over a Superdex 200 Increase equilibrated with 25 mM HEPES, 100 mM NaCl, 5 mM DM, pH 7.0 (functional assays) or 10 mM HEPES, 100 mM NaF, 5 mM DM, pH 7.0 (crystallography). Most mutants express to levels comparable to WT, and all run as monodisperse homodimers on gel filtration. Liposomes reconstituted with Ec2 were formed by mixing purified channels with 10 mg/mL stocks of EPL that had been dried down, washed and dried once with pentane, and solubilized with 30 mM CHAPS in 50 mM Tris, 100 mM NaCl, pH 7.5. Final protein concentration was 0.2 μg protein/ mg lipid, so that roughly half of the liposomes contained only a single channel and the rest were protein-free (*Walden et al., 2007*). Detergent was removed via dialysis using 10 kDa MWCO Slide-A-Lyzer (ThermoFisher [Waltham, MA]) cassettes against 3x400 mL per 10 mg lipid of 25 mM HEPES-NaOH pH 7.0 with 300 mM of either KF or KCl. Each round of dialysis was at least 6 h at RT. Final liposomes were freeze-thawed 3x prior to use in functional studies.

Monobody S9 was expressed from the pHFT2 in BL21(DE3) cells by growing cells to OD 0.8 at 37 C, transferring to 30 C, and inducing with 0.2 mM IPTG for 3 hr. Cell pellets were resuspended in 50 mM Tris, 100 mM NaCl, pH 7.5, cells were broken by sonication, lysate was clarified by centrifugation, and monobody was bound to Talon cobalt resin by batch binding for 3 hr at RT. After binding and washing with 20 mM Tris, 100 mM NaCl, pH 7.5, TEV protease (0.2 mg / L of cell culture) was added to the resin to cleave the TEV site located after the N-terminal His-tag. Cleavage ran overnight at RT, at which point cleaved monobody was rinsed off the column, concentrated, and purified over a Superdex 75 increase gel filtration column equilibrated with 10 mM HEPES, 100 mM NaF, pH 7.0.

## Channel activity assays

Channel activity was assessed by either a liposome-based anion efflux method (*Stockbridge et al., 2013*) or by single-channel recording of $F^-$ currents in planar phospholipid bilayers. For anion efflux measurements, liposomes were first extruded by 21 passes through a 0.4 μm membrane filter. 100 μL of extruded liposomes were centrifuged through a 1.5 mL G-50 Sephadex column loaded with 300 mM K-isethionate, 25 mM HEPES-NaOH pH 7.0 and diluted 20-fold into a stirred chamber with 3.8 mL flux buffer (300 mM K-isethionate, 1 mM NaF, 25 mM HEPES-NaOH pH 7.0). Liposomes for chloride efflux were diluted 10-fold into similar efflux buffer with 1 mM NaCl. Halide concentration in the chamber was monitored continuously with a $F^-$ or $Cl^-$-specific electrode. Efflux was initiated by adding 1 μM valinomycin (Vln), and total trapped $F^-/Cl^-$ was determined by adding 30 mM β-octyl-glucoside at the end of the run. The flux assay's dynamic range is limited by the electrode time response (~1 s), equivalent to single-channel turnover rates below ~30,000 ions/sec, roughly 4% of the WT single-channel current under similar conditions (*Turman et al., 2015*). All efflux experiments were repeated 4–8 times with mutant proteins from at least 2 independent purifications. Single-channel recording was done at 200 mV holding potential with a Nanion (Munich, Germany) mini-Orbit system as described (*Last et al., 2016*), with 300 mM NaF, 10 mM NaCl, 15 mM MOPS-NaOH pH 7.0 on both sides of the bilayer, in the presence of 150 nM monobody S9. The monobody binds reversibly to the Fluc channel, completely blocking the current and allowing measurement of the zero-current level. Average channel currents derived from at least 4 independent single-channel measurements.

## Crystallography

Crystal structures of various Fluc mutants in complex with a crystallization chaperone, monobody S9 (*Stockbridge et al., 2013*) were solved. Purified Fluc and monobody were mixed at a molar ratio of 1:1.2, with a final protein concentration of ~14 mg/mL total protein. Crystals were formed in sitting drops at 22°C by mixing protein 1 μL: 1 μL with well solution: 50 mM LiNO₃, 100 mM ADA pH 6.0–6.5, and PEG 550mme or 600 (30–31%). Fluoride in the crystal wells came from the protein gel filtration buffer, described above. Datasets were collected at the Advanced Light Source beamline 8.2.1 and 8.2.2 and were integrated and scaled using Mosflm (*Battye et al., 2011*) and Aimless (*Evans, 2011*) or Xia2/XDS/Aimless (*Kabsch, 2010*; *Winter, 2010*; *Winn et al., 2011*). Phases were determined by molecular replacement in PHASER (*McCoy et al., 2007*) using F80I Ec2-S9 (PDB #5KBN) as search model unless otherwise stated. Phenix (*Adams et al., 2010*) and Refmac5

**Table 2.** Crystallographic data collection and refinement statistics

| Data collection | F80Y<br>PDB 6B24 | F80M<br>PDB 6B2A | F83M<br>PDB 6B2B | T114S<br>PDB 6B2D |
|---|---|---|---|---|
| Spacegroup | $P4_1$ | $P4_1$ | $P4_1$ | $P4_1$ |
| Cell dimensions | | | | |
| a, b, c (Å) | 88.4, 88.4, 146.1 | 87.3, 87.3, 143.2 | 87.0, 87.0, 141.4 | 87.1, 87.1, 143.9 |
| a, b, g (°) | 90, 90, 90 | 90, 90, 90 | 90, 90, 90 | 90, 90, 90 |
| Resolution (Å) | 42.7–2.75 (2.90–2.75) | 43.7–2.65 (2.78–2.65) | 47.1–2.60 (2.72–2.60) | 37.8–3.01 (3.19–3.01) |
| $R_{meas}$ | 0.111 (1.40) | 0.099 (1.52) | 0.107 (1.98) | 0.106 (1.57) |
| I/$\sigma$ | 11.2 (1.5) | 14.4 (1.5) | 15.0 (1.5) | 14.1 (1.5) |
| $CC_{1/2}$ | 0.998 (0.714) | 0.997 (0.667) | 1.00 (0.662) | 0.999 (0.707) |
| Completeness | 95.9 (97.2) | 100 (100) | 99.9 (99.9) | 99.8 (99.6) |
| Multiplicity | 10.1 (9.5) | 9.9 (8.7) | 12.9 (13.2) | 8.7 (8.9) |
| Refinement Statistics | | | | |
| Resolution (Å) | 42.7–2.75 | 43.7–2.65 | 47.1–2.60 | 37.8–3.01 |
| No. Reflections | 26553 | 29629 | 30718 | 20151 |
| $R_{work}$/$R_{free}$ | 0.225/0.251 | 0.225/0.243 | 0.221/0.250 | 0.238/0.254 |
| Ramachandran Favored | 0.97 | 0.97 | 0.97 | 0.97 |
| Ramachandran Outliers, % | 0 | 0 | 0.2 | 0 |
| RMS deviations | | | | |
| Bond Lengths (Å) | 0.0076 | 0.0072 | 0.0070 | 0.0065 |
| Bond Angles (°) | 1.10 | 1.12 | 1.13 | 1.03 |

DOI: https://doi.org/10.7554/eLife.31259.020

(*Winn et al., 2003*) were used for refinement, with final refinement done in Refmac. COOT (*Emsley et al., 2010*) was used for real-space refinement, and geometry validation was checked using Molprobity (*Chen et al., 2010*). Dataset and refinement statistics are in *Table 2*.

## Acknowledgements

We thank Tania Shane and Ludmila Kolmakova-Partensky for assistance with protein preps, and Dr Dennis Dougherty for insights into electrostatics of functional groups. This research used resources of the Advanced Light Source, a DOE Office of Science User Facility, contract DE-AC02-05CH11231; we are grateful for expert help and advice in data collection from the scientists managing beamline 8.2.1/8.2.2.

## Additional information

### Funding

| Funder | Grant reference number | Author |
|---|---|---|
| Howard Hughes Medical Institute | | Nicholas B Last<br>Christopher Miller |
| National Institute of General Medical Sciences | NIH GM107023 | Senmiao Sun<br>Minh C Pham |

The funders had no role in study design, data collection and interpretation, or the decision to submit the work for publication.

## Author contributions

Nicholas B Last, Conceptualization, Data curation, Formal analysis, Supervision, Validation, Investigation, Methodology, Writing—original draft, Writing—review and editing; Senmiao Sun, Data curation, Formal analysis, Validation, Investigation, Writing—original draft; Minh C Pham, Data curation, Formal analysis, Validation, Investigation, Methodology; Christopher Miller, Conceptualization, Data curation, Formal analysis, Supervision, Funding acquisition, Validation, Investigation, Methodology, Writing—original draft, Project administration, Writing—review and editing

## Author ORCIDs

Christopher Miller http://orcid.org/0000-0002-0273-8653

## Decision letter and Author response

Decision letter https://doi.org/10.7554/eLife.31259.022
Author response https://doi.org/10.7554/eLife.31259.023

## Additional files

### Supplementary files

• Transparent reporting form
DOI: https://doi.org/10.7554/eLife.31259.021

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
