## [Decision Letter]

Thank you for submitting your article "Molecular determinants of permeation in a fluoride-specific ion channel" for consideration by *eLife*. Your article has been reviewed by three peer reviewers, and the evaluation has been overseen by Baron Chanda as the Reviewing Editor and Richard Aldrich as the Senior Editor. The reviewers have opted to remain anonymous.

The reviewers have discussed the reviews with one another and the Reviewing Editor has drafted this decision to help you prepare a revised submission.

Summary:

In this manuscript the authors investigate the chemical interactions underlying ion recognition by Fluc-Ec2, a fluoride channel from *E. coli*. It is a logical continuation of earlier work from the Miller lab centered on characterization of bacterial ion channels with high selectivity for fluoride ion over chloride. Despite having the crystal structures of Fluc, the fluoride ion path in the pore remained unclear. Last and co-workers selected eight conserved residues along the hypothetical ion pathway, performed an extensive mutagenesis study (30 new mutants of Fluc-Ec2), identified which mutations perturb fluoride transport using a liposome assay and in-planner lipid bilayers recordings, and obtained four crystal structures of various mutants in an attempt to explain their observations. The most striking result is that Fluc remains active after mutation of conserved F80 residue to M but not to Y, H or W. In contrast, mutations of conserved F83 to any residue including M makes Fluc inactive. The crystal structures reported are for two inactive mutants, F80Y and F83M, and for one active F80M. All three structures have fluoride ions bound similar to WT, in contrast to a previous report where F80I and F83I lost the ability to coordinate fluoride ions. Another surprise is that mutations of "polar track" residues have unpredictable effect on channel activity. H106, which varies in Fluc alignment to nearly any polar residue, cannot be mutated even to S or N without perturbing channel activity. Mutation of similar polarity T114S made Fluc inactive, while hydrophobic substitutions of T114V and T114I yielded fully functional channels. The crystal structure of inactive T114S mutant looks identical to WT and the authors suggest the importance of T114 in stabilization of the conducting structure but not fluoride coordination during transit.

The manuscript is well written, the data is of the highest quality and the authors are thorough in their investigation of the mutants, determining crystal structures of several of the mutants and their effects on F- binding. The study uncovers several unexpected features and raises new questions about the mechanism of unusually high ion selectivity in these channels. It is very surprising, for instance, that none of the mutants altered the effective selectivity for F- over Cl^-^ but many essentially eliminated the ability of the channel to conduct F-. One would typically expect to observe alterations in some channel property beyond this all-or-nothing effect for residues that line an ion conduction pore. Despite the fact that a clear picture does not yet emerge for the chemical requirements for F- permeation and selectivity, this study is the necessary starting point for further mechanistic studies and is worthy of publication.

Essential revisions:

1) The Materials and methods section pertaining to the functional studies is lacking sufficient detail. Details are needed regarding protein purification (buffers, detergents, methods). How are the lipids handled? (Dry down, re-suspend what buffer, sonicate? Detergent added to solubilize the lipids?). What dialysis membrane, temperature, dialysis volume, time? While many of the methods may be similar to Last et al. 2016, the methods of that publication, while slightly more detailed, could use clarification along the same lines.

2) The most confusing part of the paper is the phenylalanine mutants. It seems that there are three possible scenarios: 1) the protein transports fluoride, crystal structure has ion in the pore (WT and F80M); 2) protein does not transport fluoride, crystal structure has no ion in the pore (F80I and F83I, previous work, not discussed in current paper); and 3) protein does not transport fluoride, crystal structure has ion in the pore (F80Y and F83M). It suggests that some mutations do not let fluoride to get into the pore, and other mutations bind fluoride too tightly which blocks the pore. Some comments on that in the Discussion can be helpful in projecting the bigger picture especially in comparison to alignment showing naturally occurring M and W at the position equivalent to F83. It also seems important to mention in the Discussion the previous results with Fluc-Bpe (homolog from Bortedella pertusis) where Stockbridge et. al showed equivalent single mutants F82I and F85I transporting fluoride moderately well in liposome assays. It suggests that studies with different representatives from this divergent family of proteins can give unexpectedly different results.

---

## [Author Response]

Essential revisions:1) The Materials and methods section pertaining to the functional studies is lacking sufficient detail. Details are needed regarding protein purification (buffers, detergents, methods). How are the lipids handled? (Dry down, re-suspend what buffer, sonicate? Detergent added to solubilize the lipids?). What dialysis membrane, temperature, dialysis volume, time? While many of the methods may be similar to Last et al. 2016, the methods of that publication, while slightly more detailed, could use clarification along the same lines.

We agree that the section was perhaps a bit too bare-bones. The Materials and methods section has been significantly fleshed out.

2) The most confusing part of the paper is the phenylalanine mutants. It seems that there are three possible scenarios: 1) the protein transports fluoride, crystal structure has ion in the pore (WT and F80M); 2) protein does not transport fluoride, crystal structure has no ion in the pore (F80I and F83I, previous work, not discussed in current paper); and 3) protein does not transport fluoride, crystal structure has ion in the pore (F80Y and F83M). It suggests that some mutations do not let fluoride to get into the pore, and other mutations bind fluoride too tightly which blocks the pore. Some comments on that in the Discussion can be helpful in projecting the bigger picture especially in comparison to alignment showing naturally occurring M and W at the position equivalent to F83. It also seems important to mention in the Discussion the previous results with Fluc-Bpe (homolog from Bortedella pertusis) where Stockbridge et. al showed equivalent single mutants F82I and F85I transporting fluoride moderately well in liposome assays. It suggests that studies with different representatives from this divergent family of proteins can give unexpectedly different results.

We agree that we could do more do clarify and guide the reader through this section, though we are hesitant to engage in too grand of hypothesizing given the uncertainty that remains. We have added several statements (spread across three Discussion paragraphs) and referenced the F80I/F83I results in the Discussion in order to provide a broader perspective to the results. We have also posited some (hypothetical) explanations. We also expanded and clarified our discussion of the heterodimers and the observed non-canonical residues.

We believe that an additional focus on the Bpe homologue would not expand the discussion, and may make it less clear. Although the Bpe Phe→Ile constructs still permitted some flux, the flux was severely inhibited (>100-fold), and thus we believe those results, while slightly different in magnitude, are not broadly divergent from the ones described here for Ec2.